# A Method for Using Video Presentation to Increase Cortical Region Activity during Motor Imagery Tasks in Stroke Patients

**DOI:** 10.3390/brainsci13010029

**Published:** 2022-12-23

**Authors:** Kengo Fujiwara, Rikako Shimoda, Masatomo Shibata, Yoshinaga Awano, Koji Shibayama, Toshio Higashi

**Affiliations:** 1Medical Corporation Zeshinkai Nagasaki Rehabilitation Hospital, Ginya, Nagasaki 850-0854, Japan; 2Graduate School of Biomedical Sciences, Nagasaki University, Sakamoto, Nagasaki 852-8520, Japan; 3Medical Corporation Zeshinkai Home Rehabilitation Center Ginya, Ginya, Nagasaki 850-0854, Japan

**Keywords:** motor imagery vividness, inverse video, near-infrared spectroscopy, stroke

## Abstract

Previous studies have reported that stroke patients have difficulty recalling the motor imagery (MI) of a task, also known as MI vividness. Research on combining MI with action observation is gaining importance as a method to improve MI vividness. We enrolled 10 right-handed stroke patients and compared MI vividness and cortical activity under different presentation methods (no inverted image, inverted image of another individual’s hand, and an inverted image of the patient’s nonparalyzed hand) using near-infrared spectroscopy. Images of the nonparalyzed upper limb were inverted to make the paralyzed upper limb appear as if it were moving. Three tasks (non inverted image, AO + MI (other hand), AO + MI (own hand)) were randomly performed on 10 stroke patients. MI vividness was significantly higher when the inverted image of the nonparalyzed upper limb was presented compared to the other conditions (*p* < 0.01). The activity of the cortical regions was also significantly enhanced (*p* < 0.01). Our study highlights the potential application of inverted images of a stroke patient’s own nonparalyzed hand in mental practice to promote the motor recovery of stroke patients. This technique achieved higher levels of MI vividness and cortical activity when performing motor tasks.

## 1. Introduction

Many interventions for motor paralysis of the upper limbs following stroke are currently in use, including motor therapy, functional electrical stimulation, mental practice (MP), and action observation (AO) [1,2,3,4,5]. MP is the continuous repetition of motor imagery (MI) to improve the performance of motor tasks. MI has been described as a “mental simulation” of exercise without actual body movement [6,7,8]; MP is defined as a repetitive practice of MI, with endless opportunities for practice in a safe and cost-effective manner [9]. A systematic review revealed that MP interventions resulted in increased functionality in stroke patients with post-stroke motor paralysis [3]. A Meta-analysis, and review studies have revealed that there are similar brain areas that activate both MI and actual movements, such as the premotor area (PMA), supplementary motor area (SMA), inferior parietal lobule, superior parietal lobule, cerebellum, and prefrontal cortex [10,11]. The regions and degree of brain activity differ between motor imagery in the first-person perspective and third-person perspective [12]. However, the premotor, supplementary motor, primary motor, primary somatosensory, superior parietal, and inferior parietal lobes are activated during first-person motor imagery of the upper limb similar to the brain activity observed during actual motor execution [13]. To achieve more effective MP, participants need to be able to vividly imagine their own movements [2].

Previous neurophysiological studies on how to increase MI vividness have reported that performing MI using the same posture as the task and while grasping an object used in the task influenced MI by providing the intrinsic sensory information and increasing the excitability of corticospinal tracts during MI [14,15]. In addition, performing MI while viewing a video footage, of the task, also known as action observation (AO), was also shown to increase MI vividness [16]. This AO + MI intervention also increased activity in motor-related areas such as the supplementary motor cortex and primary motor cortex, more so than AO and MI alone [17]. Independent use of MI and AO has been largely effective, and there is evidence the two processes can elicit similar motor system activity changes [18]. In a study assessing the effects of video presentation on AO, the excitability of corticospinal tracts was increased when images were presented from a first-person perspective (i.e., observed from one’s own perspective) as opposed to from a third-person perspective (i.e., observed from the perspective of another person) [19]. Interestingly, the study showed that the excitability of the corticospinal tract was increased when the images of the patient’s own hand were presented in the video rather than those of another person [20]. Clinically, however, it is impossible to capture on film the movements of a patient’s paralyzed upper limb, as some patients are unable to move their paralyzed upper limb, and thus, some patients have further difficulty with motor imagery due to immobility.

Mirror therapy (MT) is a method used to create the illusion of movement of the paralyzed limb. The stroke patient sees the movement of their paralyzed limb by observing the movement of the nonparalyzed limb via a mirror placed in a box [21]. However, this intervention is limited in that the movements used are restricted to simple movements such as flexion and extension of the fingers and palm and dorsiflexion of the wrist joints. A meta-analysis indicated that combining MT with another rehabilitation method to improve upper extremity motor function post-stroke was better than relying solely on different monotherapy [22]. In recent years, MP using laterally inverted video presentation of a subject’s non-paralyzed upper limb to improve MI vividness has been effective in improving paralyzed upper limb function; this effect was maintained after therapy ended [23]. Although MP with inverse video presentation was effective in a single case intervention study, no studies have been conducted on a large number of stroke patients or have assessed cortical activity during MI processing using neurophysiological indicators, such as near-infrared spectroscopy (NIRS).

In a previous study, we compared MI vividness in healthy adults using different presentation methods of inverted images. We found that MI vividness was significantly higher when the inverted images of the patient’s own hand were presented, as opposed to inverted images of another individual’s hand or when no inverted images were presented at all [24]. As such, we hypothesized that if the movements of a stroke patient’s nonparalyzed upper limb could be filmed and inverted to produce an image of their paralyzed upper limb moving as if it were not paralyzed, MI vividness would increase and cortical activity would be enhanced. Therefore, we investigated MI vividness and cortical activity for the paralyzed upper limb of a stroke patient under different presentation conditions (no inverted image, inverted image of another person’s hand, and inverted image of the patient’s own nonparalyzed hand) using NIRS.

## 2. Materials and Methods

### 2.1. Participants

The characteristics of the study participants are shown in Table 1. We included 10 right-handed stroke patients (males: 4, females: 6, mean age 62.6 ± 12.0 years) who all provided informed written consent. Five patients had right hemiplegia and 5 patients had left hemiplegia.

The participants underwent the Edinburgh Handedness Test [25] and were all established as being right-handed. The tasks were designed to be performed with either the left or right hand so that the task was not affected by paralysis of the dominant hand. Mean blood pressure and pulse rate before the NIRS measurements were as follows: systolic blood pressure 123.2 ± 11.0 mmHg, diastolic blood pressure 77.2 ± 8.4 mmHg, and pulse rate 71.0 ± 10.8 beats per minute. The participants’ upper limb function on the paralyzed side scored 28.4 ± 20.2 points on the Fugl-Meyer Assessment (FMA), 0.2 ± 0.6 points on the Motor Activity Log (MAL) Amount of Use (AOU), 0.2 ± 0.5 points on the Quality of Movement (QOM), and 14.8 ± 21.1 points on the Action Research Arm Test (ARAT). The Fugl-Meyer Assessment (FMA) is a measure of motor paralysis function in the upper extremity. The Motor Activity Log (MAL) Amount of Use (AOU), 0.2 ± 0.5 points on the Quality of Movement (QOM), are quantity and quality measures, respectively, of how well the paralyzed upper extremity is used in daily life activities. The Action Research Arm Test (ARAT) is a measure of the ability of a paralyzed upper limb to carry items. A higher score indicates milder motor paralysis.

Cognitive function was assessed through the Mini-Mental State Examination (MMSE) which resulted in a score of 28.9 ± 1.1; it was assumed that the participants understood the purpose and protocol for the experiment and were able to perform it.

Exclusion criteria included individuals with a history of chronic stroke, dementia, or other neurological disease, respiratory disease, blood pressure fluctuation in the end-sitting position, inability to perform MI tasks due to higher brain dysfunction (MMSE score less than 24), and no motor paralysis in the upper limb.

This study was conducted with the approval of the Nagasaki Rehabilitation Hospital Ethics Review Committee (Approval No. R1-09). The participants were given a full explanation of the research and were asked to sign a consent form before participating in the study.

### 2.2. MI Task

The MI task consisted of a first-person viewpoint of reaching forward with the paralyzed upper limb to grasp a cup on a desk. Participants used their non-paralyzed limbs to complete tasks, and then we inverted the image to simulate the movement of the paralyzed limbs. This was performed while in a seated position without moving the trunk. The movement proceeded with lifting the cup, returning the cup to the place on the desk, releasing the hand, and returning the paralyzed upper limb to its starting position in front of the trunk. The speed of this carrying motion was calculated from the starting position of the upper limb and was determined as 1 s for each of the following four actions: reaching from the starting position, grasping and lifting the cup, placing the cup on the spot, and returning the limb to the starting position. A metronome was used to indicate the rhythm of one movement per second (1 Hz) [26] so that the speed of the carrying motion could be standardized (Figure 1). The patient was provided with repeated practice sessions to ensure that the carrying motion could be executed correctly with the nonparalyzed limb. When the patient was deemed ready, MI was conducted with the carrying action being performed five times. Video images of this movement were captured using a smartphone (iPhone XS, Apple, USA). This video was edited using free application software (SymPlayer, Masayo Tachikawa, Japan and iMovie, Apple, USA) to create a video that appeared as though the participant was performing the motion with the paralyzed upper limb. This was subsequently presented during the MI task of the paralyzed upper limb (Figure 2).

Two inverted images were created: one with the participant’s hand and the other with the hand of another person. To differentiate between the participants’ hands and the hands of others, images of hands that appeared younger than the participants’ were prepared so that they could be easily distinguished from the patients’ hands. MI vividness was then assessed using the Visual Analog Scale (VAS) [27,28]. The patients were asked to mark either end of a 100 mm horizontal line according to MI feasibility (0 = could not do MI at all, and 100 = could do MI very easily).

Participants were given three MI tasks: an MI task without an inverted image (MI only), an MI task with an inverted image of another person’s hand (other hand AO + MI), and an MI task with an inverted image of their own hand (own hand AO + MI). The three MI tasks were performed with 5 min breaks in between, using the Rand function to avoid influencing the subject’s learning effect. NIRS was then used to measure cerebral hemodynamics during the three tasks. In these tasks, participants were instructed to perform MI from a first-person viewpoint [29] as if they were performing the task themselves, and not to contract their paralyzed upper limb muscles during the task. The subjective MI vividness was assessed immediately after the MI tasks were completed using the recorded NIRS measurements. The experimental protocol is shown in Figure 3.

### 2.3. Near-Infrared Spectroscopy

NIRS measurements were performed using an optical topography system (ETG4000, Hitachi Medical Corporation, Japan) in a private evaluation room (room temperature: 24 °C) at the Nagasaki Rehabilitation Hospital to avoid introducing stimuli from the surrounding environment. The measurement was taken in a seated position with both upper limbs on the table and a cup placed within maximum reach of the patient.

NIRS probes were positioned at the Cz position (midpoint of the crown of the head) according to the international 10–20 method in a 4 × 4 probe set (Figure 4) [30]. Near-infrared light (625 and 830 nm) with high biological transparency was emitted from the NIRS probes and irradiated over the scalp to measure relative changes in near-infrared light absorption. These values were converted to changes in oxygenated hemoglobin (oxy-Hb) and deoxygenated hemoglobin (deoxy-Hb) concentrations based on the modified Beer-Lambert law [31,32,33]. There is no reported difference in optical path length between the left and right target areas for this NIRS probe [31].

The regions of interest (ROI) for this study were the left and right sensorimotor cortices (sensorimotor cortex: SMC), premotor area (PMA), prefrontal cortex (PFC), pre-supplementary motor area (pre-SMA), and supplementary motor area (SMA) (Figure 4). Based on previous studies, we designated the channels as follows: (1) 18 and 22 ch as left-SMC, (2) 21 and 24 ch as right-SMC, (3) 9, 12, 13, and 16 ch as SMA, (4) 2, 5, and 6 ch as pre-SMA, (5) 8, 11, and 15 ch as left-PMA, (6) 10, 14, and 17 ch as right-PMA, (7) 1 and 4 ch as left-PFC, and (8) 3 and 7 ch as right-PFC [34,35,36,37].

The NIRS measurements were performed in a block design [37] with three consecutive cycles of 20 s of MI tasks alternating with 30 s of rest for each condition. Since oxy-Hb has been reported to be a more accurate indicator of activation than deoxy-Hb [38], the change in oxy-Hb concentration during the MI task was used as an index of regional cerebral hemodynamics. More importantly, it has been reported that the NIRS and functional magnetic resonance imaging (fMRI) of cerebral blood flow dynamics in the cortical region during MI tasks are similar, and that cutaneous blood flow has no effect on cerebral blood flow [39]. The data obtained were analyzed for changes in oxy-Hb concentration in each region in integral mode, which is calculated by averaging three cycles of data. As for the baseline, the data were averaged over the 5 s immediately prior to the start of the MI task and the 5 s following its completion [40,41]. The data recorded 5 s following the start of the MI task until 20 s following its completion (15 s of data) were used [42] (Figure 4), which takes into consideration the time required for cerebral blood flow to increase in response to neural activity. The oxy-Hb values for each region were converted to Z-scores. If obvious artifacts were observed, they were removed from the waveform, and the average waveform was calculated via integral analysis.

### 2.4. Statistical Analysis

Statistical software (Statistical Package for the Social Sciences [SPSS] version 22.0, International Business Machines [IBM], United States) was used for the statistical analyses. A repeated measurement one-way analysis of variance (ANOVA) was conducted using the Z-score of each ROI (PFC, PMA, Pre-SMA, SMA, SMC) during MI and the three conditions of the MI task (MI only, other hand AO + MI, and own hand AO + MI) and the main factor. ANOVA was performed for the VAS of the different video presentation methods (MI only, other hand AO + MI, and own hand AO + MI). In all cases, Bonferroni’s post hoc test was employed. All significance levels were set to less than 0.05.

## 3. Results

The statistical analysis of repeated measures ANOVA revealed no interaction between the Z-score of each ROI (PFC, PMA, Pre-SMA, SMA, or SMC) during the MI and the three conditions of the MI task (MI only, other hand AO + MI, and own hand AO + MI), indicating a main effect of the three MI task conditions (F_(1,2)_ = 5.393, *p* = 0.018) (MI only, other hand AO + MI, and own hand AO + MI).

### 3.1. Activity of Cortical Regions during MI Task

All ROIs were activated during the MI for all three tasks (Figure 5). The Z-score for the ROI was compared for the three conditions. As shown in Figure 6, cortical activity was increased for all three MI task conditions, but no interaction effect was observed. Although not statistically significant, oxy-Hb was higher in the SMA, left-PMA, right-PMA, left-SMC, and right-SMC ROIs in the case of own hand AO + MI. Therefore, we subsequently conducted a comparison of oxy-Hb Z-scores using Bonferroni’s post hoc test and determined that the score was significantly higher for the patient’s own hand AO + MI compared with that of MI only or other hand AO + MI (F_(1,2)_ = 37.327, *p* = 0.000). (*p* < 0.01) (Figure 6).

### 3.2. MI Vividness

MI vividness was evaluated using the VAS following NIRS measurements, and the effect observed showed a mean of 48.2 ± 9.7 mm for MI only, 56.7 ± 13.3 mm for another hand AO + MI, and 82.2 ± 11.2 mm for own hand AO + MI. Furthermore, Bonferroni’s post hoc test demonstrated that having a visual representation of a hand increased the MI vividness significantly compared the use of the patient’s own hand (own hand AO + MI) compared with MI only. (F_(1,2)_ = 5.846, *p* = 0.006) (*p* < 0.01) (Figure 7).

## 4. Discussion

For this study, images of the patient’s own nonparalyzed hand while performing a specific MI task were inverted, and then presented so that it appeared as if the patient’s paralyzed hand was actually performing the same movement. Interestingly, our study of stroke patients yielded results similar to those from previous studies performed with healthy adults. In the comparison of MI vividness in healthy adults using different presentation methods of inverted images, MI vividness was significantly higher with the presentation of the inverted image of the patient’s own hand as opposed to using an inverted image of another person’s hand, or when no inverted image was used [24]. Furthermore, a previous study demonstrated a positive correlation between MI vividness and corticospinal tract excitability during MI tasks in healthy adults [43]. 

With regard to image presentation, the excitability of corticospinal tracts has been reported to be higher when first-person perspective was used (i.e., observing the hand from one’s own viewpoint) compared to third-person perspective (i.e., objectively observing oneself from the viewpoint of another person) [20]. In addition, it has been reported that the excitability of corticospinal tracts increased when the participant’s own hand was presented in the video [21] compared to another person’s hand. Furthermore, an NIRS study comparing differences in presentation methods in healthy adults showed that the oxy-Hb in cortical areas was significantly higher when the inverted image of the participant’s hand was presented as opposed to when the inverted image of another’s hand was shown, or when no inverted image was used at all [24]. This study supports our results which demonstrated significantly higher oxy-Hb values across all ROIs when inverted images of a stroke patient’s own hand were presented compared to when no image was used. Based on these results, we believe that the use of inverted images of the patient’s hand from a first-person perspective may increase cortical activity during MI in stroke patients.

In a single-case intervention report, MP was performed with chronic stroke patients by capturing images of the movement of the nonparalyzed upper limb and subsequently presenting inverted images to represent movement of the paralyzed upper limb. Significant improvement was observed in upper limb function and MI vividness [23]. Furthermore, MI can be performed early during the recovery stage of motor paralysis in stroke patients, even in the state of flaccid paralysis [44,45]. There have also been reports of improved performance with AO + MI for the lower extremities [46]. Recently, MI has been used in the brain-machine interface (BMI), and it has been demonstrated that NIRS is able to detect changes in brain activity during MI and motor execution of upper limb movements similar to fMRI [47,48]. Based on these reports and the results of this study, we believe that the use of inverted images of the patient’s hand in MP for stroke patients highlights its potential application during rehabilitation to promote the recovery of motor paralysis, as this technique is able to maintain high levels of MI vividness in addition to increasing cortical activity. In this study, we did not have a control group, so we were not able to examine differences between healthy subjects and stroke patients in a similar MI task. However, since stroke patients may have difficulty with motor imagery due to immobility caused by motor paralysis, we believe that the differences in MI vividness and cortical activity between the different video images in this study may be more relevant for clinical application. Therefore, in future studies, it is necessary to verify the effectiveness of the MP intervention by using an inverted AO +MI (own hand) approach in stroke patients.

There were several limitations to this study that should also be taken into consideration. First, there was no control group. We believe that future studies should also compare differences between stroke patients and healthy subjects as a control group. Further, due to the small sample size used, the analysis could not be performed based on the classification of cerebral hemorrhage/stroke, injury site, or by right hemiplegia/left hemiplegia. Second, while the chosen MI task made use of a cup as the practice object, it was not possible to capture tasks that would be more difficult to perform with the nonparalyzed upper limb, and as such, not all MI tasks are likely to result in the same cortical area activity as that observed in this study. Finally, there was no interaction between MI only, other hand AO + MI, own hand AO + MI, and each ROI; therefore, it is not clear which cortical regions became increasingly activated under the own hand AO + MI condition.

## 5. Conclusions

The results of this study showed that the MI vividness and cortical activity in the paralyzed upper limb of a stroke patient were significantly higher when the subject’s own hand was inverted than when no inverted image was shown. The results of this study suggest that MI vividness can be enhanced and cortical activity during MI can be activated by presenting the subject’s own hand inverted image when practicing more effective MP. Our future studies will examine effective video presentation methods for practicing MP in stroke patients, using healthy subjects as a control group. The intervention effects on the paralyzed upper limb function should be evaluated when practicing MP using a video presentation method that can increase MI vividness in stroke patients.

## Figures and Tables

**Figure 1 brainsci-13-00029-f001:**
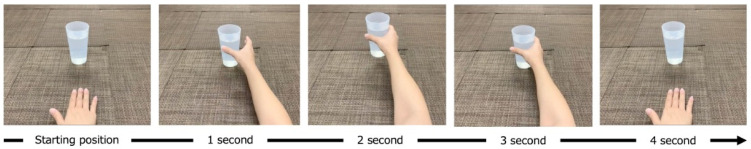
Motor imagery (MI) task of the paralyzed upper limb grasping a cup on a desk. The MI task consisted of a first-person viewpoint of reaching forward with the paralyzed upper limb to grasp a cup on a desk. This was performed while in a seated position without moving the trunk. Then, the movement proceeds with lifting the cup, returning the cup to the place on the desk, releasing the hand, and returning the paralyzed upper limb to its starting position in front of the trunk.

**Figure 2 brainsci-13-00029-f002:**
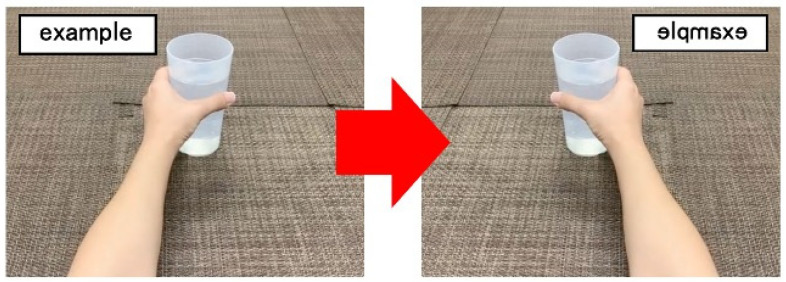
Inverted MI task images. We took images of the movements of a stroke patient’s nonparalyzed upper limb and inverted the images to make the paralyzed upper limb seem as if it were moving.

**Figure 3 brainsci-13-00029-f003:**
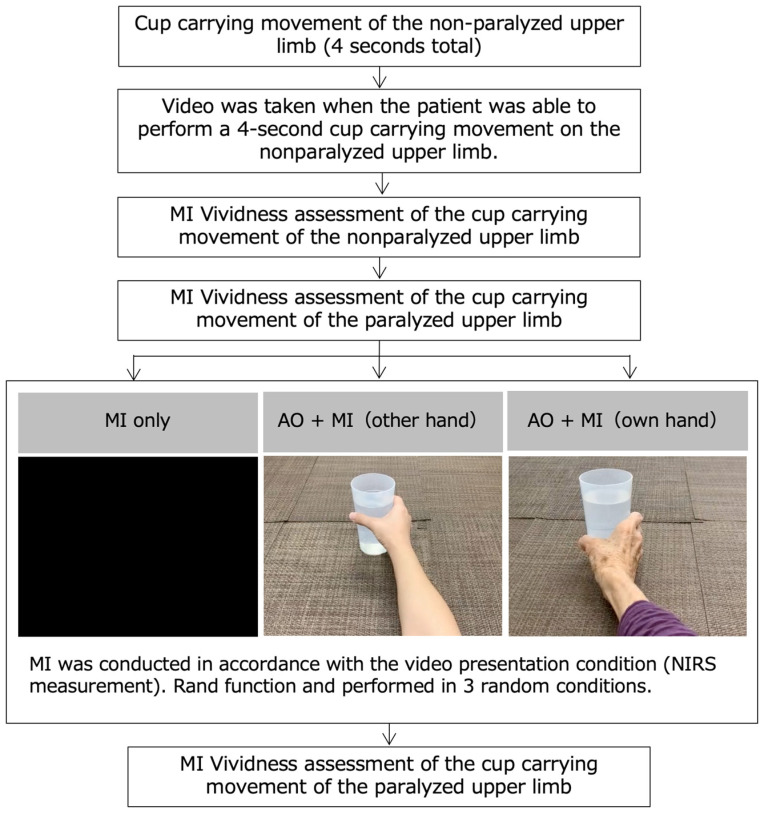
The experimental protocol. Participants were given three MI tasks: an MI task without an inverted image (MI only), an MI task with an inverted image of another person’s hand (other hand action observation [AO] + MI), and an MI task with an inverted image of their own hand (own hand AO + MI). The subjective MI vividness was assessed immediately after the MI tasks were completed using the recorded NIRS measurements.

**Figure 4 brainsci-13-00029-f004:**
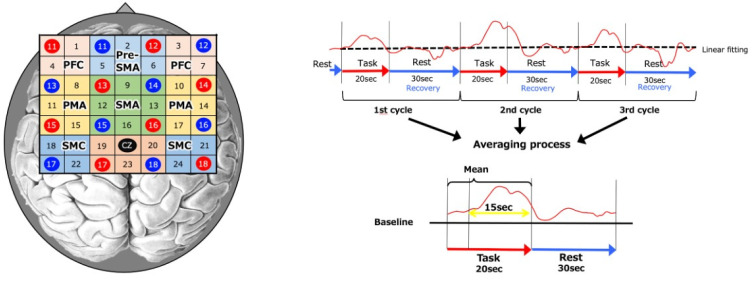
Near-infrared spectroscopy (NIRS). NIRS measurements were performed using an optical topography system. NIRS probes were positioned at the Cz position (midpoint of the crown of the head) according to the international 10–20 method in a 4 × 4 probe set. As for the baseline, the data were averaged over the 5 s immediately prior to the start of the MI task and the 5 s following its completion. The data recorded 5 s following the start of the MI task until 20 s following its completion (15 s of data) were used.

**Figure 5 brainsci-13-00029-f005:**
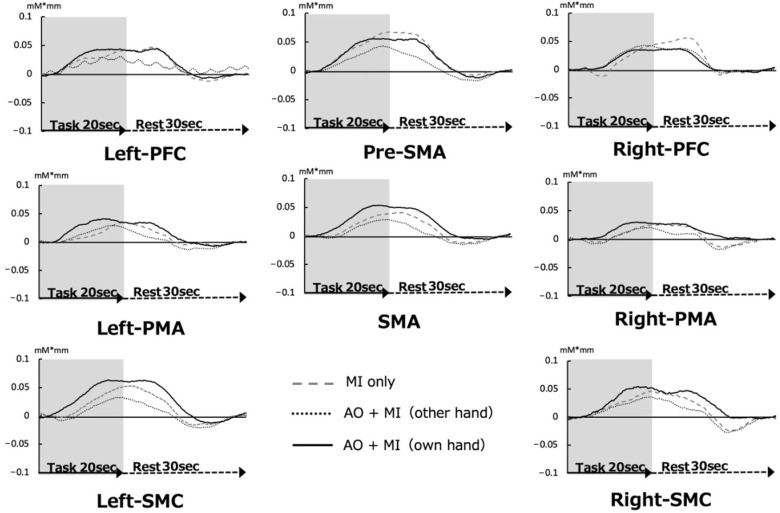
All regions of interest (ROI) during three MI tasks. Cortical activity was observed for all three MI task conditions.

**Figure 6 brainsci-13-00029-f006:**
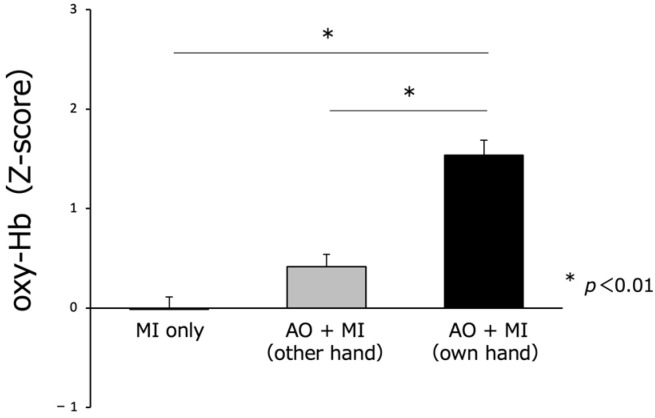
Comparison of oxygenated hemoglobin (oxy-Hb) (Z-score) during the MI of paralyzed upper limbs. Oxy-Hb in the cortical region was significantly higher in own hand AO + MI than in MI only or other hand AO + MI.

**Figure 7 brainsci-13-00029-f007:**
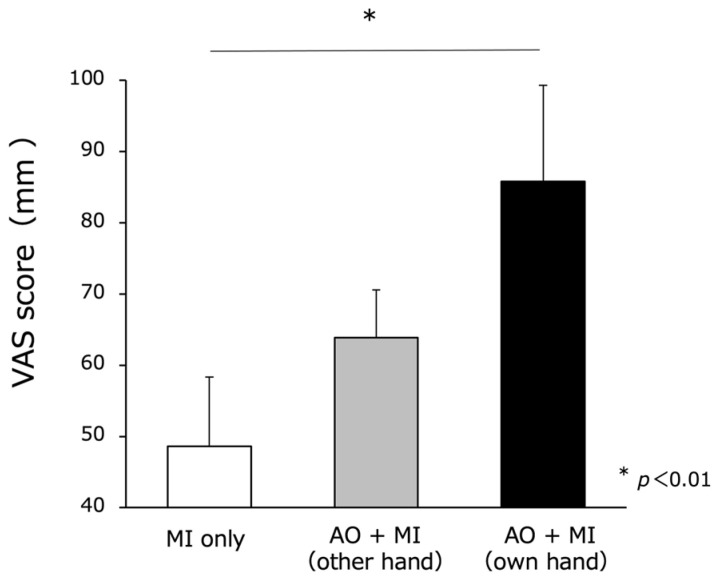
MI vividness. MI vividness was evaluated using visual analog scale (VAS) following NIRS measurements, and the effect observed showed a mean of 48.2 ± 9.7 mm for MI only, 56.7 ± 13.3 mm for other hand AO + MI and 82.2 ± 11.2 mm for own hand AO + MI.

**Table 1 brainsci-13-00029-t001:** Characteristics of the study participants.

					Blood Pressure and Pulse Rate before NIRS Measurement			Upper Limb Functional Assessment				Cognitive Function
Disease	Paralyzed Side	Gender	Age	Days from Onset to Date of Measurement	Systolic Blood Pressure	Diastolic Blood Pressure	Pulse	FMA	MAL (AOU)	MAL (QOM)	ARAT (Paralyzed Side)	MMSE
CH	R	F	59	34	120	70	65	4	0	0	0	29
CI	R	M	55	30	140	90	70	54	0.38	0.61	50	30
CH	R	F	61	55	114	84	85	53	2	1.8	57	30
CI	R	M	78	28	110	72	68	31	0.3	0.4	17	29
CI	R	F	48	35	124	80	80	4	0	0	0	26
CH	L	F	81	37	108	60	78	4	0	0	0	29
CI	L	F	79	40	138	78	51	34	0	0	10	29
CH	L	M	56	25	126	76	80	35	0	0	3	29
CI	L	M	53	54	120	84	81	49	0	0	5	30
CI	L	F	56	28	132	78	60	16	0	0	6	28
AVG			62.6	36.6	123.2	77.2	71.8	28.4	0.2	0.2	14.8	28.9
SD			12.0	10.4	11.0	8.4	10.8	20.2	0.6	0.5	21.1	1.1
SE			3.8	3.31	3.4	2.6	3.4	6.4	0.1	0.1	6.6	0.3

CH: cerebral hemorrhage. CI: cerebral infarction. R: Right. L: Left. M: Man. F: Female.

## Data Availability

Not applicable.

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
