# Peer review of "A Method for Using Video Presentation to Increase Cortical Region Activity during Motor Imagery Tasks in Stroke Patients"

_brainsci, 2022, doi:10.3390/brainsci13010029_

Round 1

Reviewer 1 Report

In this study, authors investigated the method for using video presentation to increase cortical region activity during motor imagery tasks in stroke patients. They observed use the inverted images of a stroke patient’s own non-paralyzed hand increased both MI vividness and cortical activity. The research theme of the article is very interesting and meaningful, but a few comments should be addressed to make the findings clearer and more robust.

Abstract

Ø   The methods is not clear enough, and the grouping is not clear enough. It can be consistent with the "Method" part: MI AO+MI (other hand) AO+MI (own hand)

Ø  Abstract is generally well written, but I miss some concrete data (descriptive and/or effect sizes with p-values)

Introduction

Ø  It can supplement the physiological basis related to MI

Ø  Line 27-41 The relationship between MP AO MI needs to be improved

Ø  line53 “… we compared MI vividness in healthy adults… ” and line 55 “…when the inverted images of the patient’s own hand were presented…”Since the subject is a healthy person, it seems inappropriate to use the word "patient"

Ø  The introduction does not seem to mention the difference in MI vividness between the stroke population and healthy adults. How can the innovation of the research be reflected when the research object is changed from healthy people to stroke patients?

Materials and Methods

Ø  “2.1. Participants” The paralyzed years of the subjects, which side is paralyzed

Ø  Line67 “right-handed stroke patients” But if the patient is paralyzed on the right side, will it affect their hand habits?

Ø  Line72-78What does this series of scores mean? If it can be compared with the scores of normal limbs, the data will be more meaningful.

Ø  Line97 “The MI task consisted of a first-person viewpoint of reaching forward with the paralyzed upper limb to grasp a cup on a desk.” Are authors sure the paralyzed limb of the stroke patient can complete this task? Combine line106 “The patient was provided with repetitive practice sessions to ensure that the carrying motion could be executed correctly with the nonparalyzed limb.”

What I understand is that participants use non-paralyzed limbs to complete tasks, and then invert the image to simulate the movement of paralyzed limbs. If my understanding is correct, please correct the ambiguity (line97, line117). If not, I doubt that the paralyzed limb can complete grafting a cup on a desk.

Ø  Line 203 line 205

MI task is a within-group factor, and statistical analyses of repeated measurement ANOVA should be used. Rather than a one-way analysis of variance and two-way.

Ø  Line 208

5% changed to 0.05

Results

Ø  It is recommended to report the F value, df value, and P value by the specification.

Ø  Histogram of z value of each brain region. Take the previous articles published by the author as an example DOI:10.4103/1673-5374.313058.

Ø  Whether to consider the inclusion of healthy subjects for comparative investigation of the differences between the two groups of people

Discussion

Ø  The discussion seems to be not deep enough.

Ø  If the author does not consider being included in the control group, the possible differences between normal people and paralyzed people should be discussed, and the uniqueness of the study should be emphasized.

Ø  It should be added that there is no control group in this study. (But I strongly suggest that the author add a group of healthy adult controls to improve the quality of the research.)

Conclusions

Ø  Line 316 AO+MI works best, Why the conclusion is that MI is beneficial

Ø  The conclusion is not accurate enough

Author Response

RESPONSES TO REVIEWER 1’S COMMENTS

The authors would like to thank the reviewer for his/her constructive critique aimed at improving the manuscript. We have made every effort to address the issues raised and to respond to all comments. The revisions are indicated in red font in the revised manuscript. Below is a detailed, point-by-point response to the reviewer's comments. We hope that our revisions meet the reviewer’s expectations.

Abstract

The methods is not clear enough, and the grouping is not clear enough. It can be consistent with the "Method" part: MI AO+MI (other hand) AO+MI (own hand)

Response: In accordance with the reviewer’s comment, we have added the following text to the Abstract section

<revised>

Images of the nonparalyzed upper limb were inverted to make the paralyzed upper limb appear as if it were moving. Three tasks (non-inverted image, AO+MI (other hand), AO+MI (own hand)) were randomly performed on 10 stroke patients

Abstract is generally well written, but I miss some concrete data (descriptive and/or effect sizes with p-values)

Response: As suggested, we have added more concrete data to the Abstract section as follows

<original>

Results: MI vividness and cortical activity were significantly higher when the inverted image of the nonparalyzed upper limb was presented compared to when the inverted image of the paralyzed upper limb was presented.

<revised>

Results: MI vividness was significantly higher when the inverted image of the nonparalyzed upper limb was presented compared to the other conditions (p<0.01). The activity of the cortical regions was also significantly enhanced (p<0.01).

Introduction

 It can supplement the physiological basis related to MI

Response: We have modified and added the following text to the Introduction section:

<original>

MP is the continuous repetition of motor imagery (MI) to improve the performance of motor tasks. A systematic review revealed that MP interventions resulted in increased functionality in stroke patients with post-stroke motor paralysis [3]. Previous studies have also demonstrated that MI induces activity in cortical brain regions equivalent to that during actual movement [6]. However, for effective MP, the subjects must be able to vividly imagine their movements [2].

<revised>

MP is the continuous repetition of motor imagery (MI) to improve the performance of motor tasks. MI has been described as a "mental simulation" of exercise without actual body movement [6-8]; MP is defined as a repetitive practice of MI, with endless opportunities for practice in a safe and cost-effective manner [9].A systematic review revealed that MP interventions resulted in increased functionality in stroke patients with post-stroke motor paralysis [3]. A Meta-analysis, and review studies have revealed that there are similar brain areas that activate both MI and actual movements, such as the premotor area (PMA), supplementary motor area (SMA), inferior parietal lobule, superior parietal lobule, cerebellum, and prefrontal cortex [10,11]. The regions and degree of brain activity differ between motor imagery in the first-person perspective and third-person perspective [12]. However, the premotor, supplementary motor, primary motor, primary somatosensory, superior parietal, and inferior parietal lobes are activated during first-person motor imagery of the upper limb similar to the brain activity observed during actual motor execution [13]. To achieve more effective MP, participants need to be able to vividly imagine their own movements [2].

Line 27-41 The relationship between MP AO MI needs to be improved

Response: Accordingly, we have modified and added the following text to the Introduction:

<original>

Previous neurophysiological studies on how to increase MI vividness have reported that performing MI in the same posture as the task and while grasping an object used in the task influenced MI with the intrinsic sensory information and increased the excitability of corticospinal tracts during MI [7,8]. In addition, performing MI while viewing a video footage of the task was also shown to increase MI vividness [9]. This intervention also increased activity in motor-related areas such as the supplementary motor cortex and primary motor cortex, more so than AO and MI alone [10].

<revised>

Previous neurophysiological studies on how to increase MI vividness have reported that performing MI using the same posture as the task and while grasping an object used in the task influenced MI by providing the intrinsic sensory information and increasing the excitability of corticospinal tracts during MI [14,15]. In addition, performing MI while viewing a video footage, of the task, also known as action observation (AO), was also shown to increase MI vividness [16]. This AO+MI intervention also increased activity in motor-related areas such as the supplementary motor cortex and primary motor cortex, more so than AO and MI alone [17]. Independent use of MI and AO has been largely effective, and there is evidence the two processes can elicit similar motor system activity changes [18]. In a study assessing the effects of video presentation on AO, the excitability of corticospinal tracts was increased when images were presented from a first-person perspective (i.e., observed from one's own perspective) as opposed to from a third-person perspective (i.e., observed from the perspective of another person) [19]. Interestingly, the study showed that the excitability of the corticospinal tract was increased when the images of the patient's own hand were presented in the video rather than those of another person [20]. Clinically, however, it is impossible to capture on film the movements of a patient's paralyzed upper limb, as some patients are unable to move their paralyzed upper limb, and thus, some patients have further difficulty with motor imagery due to immobility.

 line53 “… we compared MI vividness in healthy adults… ” and line 55 “…when the inverted images of the patient’s own hand were presented…”Since the subject is a healthy person, it seems inappropriate to use the word "patient"

line53 "...we compared MI vividness in healthy adults..." and line55 "...when the inverted images of the patient ‘s own hand... "

Response: Accordingly, we have modified and added the text to the Introduction section as follows

<original>

we compared MI vividness in healthy adults using different presentation methods of inverted images. We found that MI vividness was significantly higher when the inverted images of the patient’s own hand were presented, as opposed to when the inverted images of another's hand or when no inverted images were presented at all [14].

<revised>

We found that MI vividness was significantly higher when the inverted images of the patient’s own hand were presented, as opposed to inverted images of another individual’s hand or when no inverted images were presented at all [24]

The introduction does not seem to mention the difference in MI vividness between the stroke population and healthy adults. How can the innovation of the research be reflected when the research object is changed from healthy people to stroke patients?

Response: We have added the following text to the Introduction section:

<revised>

Previous neurophysiological studies on how to increase MI vividness have reported that performing MI using the same posture as the task and while grasping an object used in the task influenced MI by providing the intrinsic sensory information and increasing the excitability of corticospinal tracts during MI [14,15]. In addition, performing MI while viewing a video footage, of the task, also known as action observation (AO), was also shown to increase MI vividness [16]. This AO+MI intervention also increased activity in motor-related areas such as the supplementary motor cortex and primary motor cortex, more so than AO and MI alone [17]. Independent use of MI and AO has been largely effective, and there is evidence the two processes can elicit similar motor system activity changes [18]. In a study assessing the effects of video presentation on AO, the excitability of corticospinal tracts was increased when images were presented from a first-person perspective (i.e., observed from one's own perspective) as opposed to from a third-person perspective (i.e., observed from the perspective of another person) [19]. Interestingly, the study showed that the excitability of the corticospinal tract was increased when the images of the patient's own hand were presented in the video rather than those of another person [20]. Clinically, however, it is impossible to capture on film the movements of a patient's paralyzed upper limb, as some patients are unable to move their paralyzed upper limb, and thus, some patients have further difficulty with motor imagery due to immobility.

Mirror therapy (MT) is a method used to create the illusion of movement of the paralyzed limb. The stroke patient sees the movement of their paralyzed limb by observing the movement of the nonparalyzed limb via a mirror placed in a box [21]. However, this intervention is limited in that the movements used are restricted to simple movements such as flexion and extension of the fingers and palm and dorsiflexion of the wrist joints. A meta-analysis indicated that combining MT with another rehabilitation method to improve upper extremity motor function post-stroke was better than relying solely on different monotherapy [22]. In recent years, MP using laterally inverted video presentation of a subject’s non-paralyzed upper limb to improve MI vividness has been effective in improving paralyzed upper limb function; this effect was maintained after therapy ended [23]. Although MP with inverse video presentation was effective in a single case intervention study, no studies have been conducted on a large number of stroke patients or have assessed cortical activity during MI processing using neurophysiological indicators, such as near-infrared spectroscopy (NIRS).

Materials and Methods

2.1. Participants” The paralyzed years of the subjects, which side is paralyzed?

Response: Thank you for the question. We have added text to the Materials and Methods:

<revised>

Five patients had right hemiplegia and 5 patients had left hemiplegia.

 Line67 “right-handed stroke patients” But if the patient is paralyzed on the right side, will it affect their hand habits?

Response: We have added the following text to the Materials and Methods section:

<revised>

The participants underwent the Edinburgh Handedness Test [25] and were all established as being right-handed. The tasks were designed to be performed with either the left or right hand so that the task was not affected by paralysis of the dominant hand.

Line72-78What does this series of scores mean? If it can be compared with the scores of normal limbs, the data will be more meaningful.

Response: We appreciate the reviewer’s suggestion. We have added the following text to the Materials and Methods section.

These series of evaluations are for the paralyzed upper extremity after stroke, and normal (i.e., perfect score) scores are 66 for FMA, 5 for MAL, and 57 for ARAT. The scores represent the upper extremity function of the movement.

<revised>

The Fugl-Meyer Assessment (FMA) is a measure of motor paralysis function in the upper extremity. The Motor Activity Log (MAL) Amount of Use (AOU), 0.2 ± 0.5 points on the Quality of Movement (QOM), are quantity and quality measures, respectively, of how well the paralyzed upper extremity is used in daily life activities. The Action Research Arm Test (ARAT) is a measure of the ability of a paralyzed upper limb to carry items. A higher score indicates milder motor paralysis.

Line97 “The MI task consisted of a first-person viewpoint of reaching forward with the paralyzed upper limb to grasp a cup on a desk.” Are authors sure the paralyzed limb of the stroke patient can complete this task? Combine line106 “The patient was provided with repetitive practice sessions to ensure that the carrying motion could be executed correctly with the nonparalyzed limb.”

What I understand is that participants use non-paralyzed limbs to complete tasks, and then invert the image to simulate the movement of paralyzed limbs. If my understanding is correct, please correct the ambiguity (line97, line117). If not, I doubt that the paralyzed limb can complete grafting a cup on a desk.

Response: We appreciate the reviewer’s comment on this point. We have clarified our description in the Materials and Methods section as follows:

<original>

The MI task consisted of a first-person viewpoint of reaching forward with the paralyzed upper limb to grasp a cup on a desk.

Figure 1. Motor imagery (MI) task of the paralyzed upper limb grasping a cup on a desk. The MI task consisted of a first-person viewpoint of reaching forward with the paralyzed upper limb to grasp a cup on a desk.

<revised>

The MI task consisted of a first-person viewpoint of reaching forward with the paralyzed upper limb to grasp a cup on a desk. Participants used their non-paralyzed limbs to complete tasks, and then we inverted the image to simulate the movement of the paralyzed limbs.

Figure 1. Motor imagery (MI) task of the paralyzed upper limb grasping a cup on a desk. The Participants use non-paralyzed limbs to complete tasks, and then the image was inverted to simulate the movement of paralyzed limbs.

  Line 203 line 205

MI task is a within-group factor, and statistical analyses of repeated measurement ANOVA should be used. Rather than a one-way analysis of variance and two-way.

Response: We appreciate the reviewer’s suggestion. We have added the following to the Materials and Methods section:

<revised>

A repeated measurement one-way analysis of variance (ANOVA) was conducted using the Z-score of each ROI (PFC, PMA, Pre-SMA, SMA, SMC) during MI and the three conditions of the MI task (MI only, other hand AO+MI, and own hand AO+MI) and the main factor.

Line 208

5% changed to 0.05

Response: We have modified the text in the Materials and Methods section accordingly.

<original>

All significance levels were set to less than 5%.

<revised>

All significance levels were set to p-value less than 0.05.

Results

 It is recommended to report the F value, df value, and P value by the specification.

Response: We have added these details to the Results section as follows

<revised>

The statistical analysis of repeated measures ANOVA revealed no interaction between the Z-score of each ROI (PFC, PMA, Pre-SMA, SMA, or SMC) during the MI and the three conditions of the MI task (MI only, other hand AO+MI, and own hand AO+MI), indicating a main effect of the three MI task conditions(F(1,2)=5.393,p=0.018) (MI only, other hand AO+MI, and own hand AO+MI).

<original>

Furthermore, the post-hoc test demonstrated that having a visual representation of a hand increased MI vividness significantly compared with MI only, as did the use of the patient’s own hand (own hand AO + MI) compared with the use of another’s hand (other hand AO + MI) (p < 0.01) (Figure 5).

<revised>There was an error in the diagram, which has been corrected.

 Furthermore, Bonferroni’s post hoc test demonstrated that having a visual representation of a hand increased the MI vividness significantly compared the use of the patient’s own hand (own hand AO+MI) compared with MI only.(F(1,2)=5.846, p=0.006)(p<0.01) (Figure 7).

<original>

Therefore, we subsequently conducted a comparison of oxy-Hb Z-scores and determined that the score was significantly higher for the own hand AO + MI compared with that of MI only (Figure 7).

<revised>There was an error in the diagram, which has been corrected.

T Therefore, we subsequently conducted a comparison of oxy-Hb Z-scores using Bonferroni’s post hoc test and determined that the score was significantly higher for the patient’s own hand AO+MI compared with that of MI only or another hand AO+MI (F(1,2)=37.327, p=0.000). (p<0.01) (Figure 6).

Histogram of z value of each brain region. Take the previous articles published by the author as an example DOI:10.4103/1673-5374.313058.

Response: In our previous study, we also presented relative values for the time course rather than the Z-score. The Z-score was used for the statistical analysis, as in the previous studies.

Whether to consider the inclusion of healthy subjects for comparative investigation of the differences between the two groups of people

Discussion

 The discussion seems to be not deep enough.

If the author does not consider being included in the control group, the possible differences between normal people and paralyzed people should be discussed, and the uniqueness of the study should be emphasized.

It should be added that there is no control group in this study. (But I strongly suggest that the author add a group of healthy adult controls to improve the quality of the research.)

Response: We appreciate the reviewer’s suggestions; however, we were unable to add a control group. Accordingly, we have added the following comment to the Discussion section:

<revised>

In this study, we did not have a control group, so we were not able to examine differences between healthy subjects and stroke patients in a similar MI task. However, since stroke patients may have difficulty with motor imagery due to immobility caused by motor paralysis, we believe that the differences in MI vividness and cortical activity between the different video images in this study may be more relevant for clinical application. Therefore, in future studies, it is necessary to verify the effectiveness of the MP intervention by using an inverted AO +MI (own hand) approach in stroke patients.

There were several limitations to this study that should also be taken into consideration. First, there was no control group. We believe that future studies should also compare differences between stroke patients and healthy subjects as a control group

Conclusions

Line 316 AO+MI works best, Why the conclusion is that MI is beneficial

The conclusion is not accurate enough

Response: We have provided additional explanation in the Conclusions section.

<original>

Overall, our results suggest that the use of inverted images of the patient’s own nonparalyzed hand while performing MI is a useful method to increase MI vividness and cortical activity in stroke patients.

<revised>

The results of this study showed that the MI vividness and cortical activity in the paralyzed upper limb of a stroke patient were significantly higher when the subject's own hand was inverted than when no inverted image was shown. The results of this study suggest that MI vividness can be enhanced and cortical activity during MI can be activated by presenting the subject's own hand inverted image when practicing more effective MP. Our future studies will examine effective video presentation methods for practicing MP in stroke patients, using healthy subjects as a control group. The intervention effects on the paralyzed upper limb function should be evaluated when practicing MP using a video presentation method that can increase MI vividness in stroke patients.

Reviewer 2 Report

In this manuscript, the authors have compared MI vividness and cortical activity of 10 stroke patients under different presentation methods (no inverted image, inverted image of another person's hand, and inverted image of the patient's nonparalyzed hand) using near-infrared spectroscopy. They have shown that using inverted image of the patient's nonparalyzed hand + MI results in the best performance. The manuscript is well-written and well-organized and the topic is interesting. However, there are few points that should be addressed.

(1) You have mentioned in the introduction "Previous studies have also demonstrated that MI induces activity in cortical brain regions equivalent to that during actual movement". I don't think the induced activity during MI is "equivalent to" that during actual movement. There are some similarities but also there seems to be some differences. Please elaborate more on your claim.

(2) The fonts in tables and figures are low-resolution. Please enhance the fonts.

(3) You have mentioned "we have prepared images of hands which generally appeared much older and could be easily distinguished from the patients’ hands". However, at least in Figure 3, I don't think the "other hand" is older than "own hand". Please more elaborate on the differences between "other hand" and "own hand" that let the patient clearly differentiates it from his/her hand.

(4) Please mention the exact p-values and not just saying that p-value<0.01 or p-value<0.05.

(5) The conclusion is too brief. I suggest the authors to extend the conclusions by adding the novel findings of this study and proposing some future works to deal with its limitations.

Author Response

AUTHORS’ RESPONSES TO REVIEWER 2’S COMMENTS

The authors would like to thank the reviewer for his/her constructive critique aimed at improving the manuscript. We have made every effort to address the issues raised and to respond to all comments. The revisions are indicated in red font in the revised manuscript. Below is a detailed, point-by-point response to the reviewer's comments. We hope that our revisions meet the reviewer’s expectations.

(1) You have mentioned in the introduction "Previous studies have also demonstrated that MI induces activity in cortical brain regions equivalent to that during actual movement". I don't think the induced activity during MI is "equivalent to" that during actual movement. There are some similarities but also there seems to be some differences. Please elaborate more on your claim.

Response: We appreciate the reviewer’s suggestion. Accordingly, we have added the following details to the Introduction section:

<revised>

MI has been described as a "mental simulation" of exercise without actual body movement [6-8]; MP is defined as a repetitive practice of MI, with endless opportunities for practice in a safe and cost-effective manner [9].A systematic review revealed that MP interventions resulted in increased functionality in stroke patients with post-stroke motor paralysis [3]. A Meta-analysis, and review studies have revealed that there are similar brain areas that activate both MI and actual movements, such as the premotor area (PMA), supplementary motor area (SMA), inferior parietal lobule, superior parietal lobule, cerebellum, and prefrontal cortex [10,11]. The regions and degree of brain activity differ between motor imagery in the first-person perspective and third-person perspective [12]. However, the premotor, supplementary motor, primary motor, primary somatosensory, superior parietal, and inferior parietal lobes are activated during first-person motor imagery of the upper limb similar to the brain activity observed during actual motor execution [13]. To achieve more effective MP, participants need to be able to vividly imagine their own movements [2].

(2) The fonts in tables and figures are low-resolution. Please enhance the fonts.

Response: We have provided new figures and tables

(3) You have mentioned "we have prepared images of hands which generally appeared much older and could be easily distinguished from the patients’ hands". However, at least in Figure 3, I don't think the "other hand" is older than "own hand". Please more elaborate on the differences between "other hand" and "own hand" that let the patient clearly differentiates it from his/her hand.

Response: We appreciate the reviewer’s suggestion. We have changed text to the Materials and Methods section and the Figure 3 as follows

<original>

Two inverted images were created: one with the participant's hand and the other with the hand of another person. To differentiate between the participants’ hands and those of others, we prepared images of hands which generally appeared much older and could be easily distinguished from the patients’ hands.

<revised>

Two inverted images were created: one with the participant's hand and the other with the hand of another person. To differentiate between the participants' hands and the hands of others, images of hands that appeared younger than the participants' were prepared so that they could be easily distinguished from the patients' hands.

(4) Please mention the exact p-values and not just saying that p-value<0.01 or p-value<0.05.

Response: Accordingly, we have added the exact p-values to the Results section

<revised>

The statistical analysis of repeated measures ANOVA revealed no interaction between the Z-score of each ROI (PFC, PMA, Pre-SMA, SMA, or SMC) during the MI and the three conditions of the MI task (MI only, other hand AO+MI, and own hand AO+MI), indicating a main effect of the three MI task conditions(F(1,2)=5.393,p=0.018) (MI only, other hand AO+MI, and own hand AO+MI).

<original>

Furthermore, the post-hoc test demonstrated that having a visual representation of a hand increased MI vividness significantly compared with MI only, as did the use of the patient’s own hand (own hand AO + MI) compared with the use of another’s hand (other hand AO + MI) (p < 0.01) (Figure 5).

<revised>There was an error in the figure, which has been corrected.

 Furthermore, Bonferroni’s post hoc test demonstrated that having a visual representation of a hand increased the MI vividness significantly compared the use of the patient’s own hand (own hand AO+MI) compared with MI only.(F(1,2)=5.846, p=0.006)(p<0.01) (Figure 7).

<original>

Therefore, we subsequently conducted a comparison of oxy-Hb Z-scores and determined that the score was significantly higher for the own hand AO + MI compared with that of MI only (Figure 7).

<revised>There was an error in the diagram, which has been corrected.

 Therefore, we subsequently conducted a comparison of oxy-Hb Z-scores using Bonferroni’s post hoc test and determined that the score was significantly higher for the patient’s own hand AO+MI compared with that of MI only or another hand AO+MI (F(1,2)=37.327, p=0.000). (p<0.01) (Figure 6)

(5) The conclusion is too brief. I suggest the authors to extend the conclusions by adding the novel findings of this study and proposing some future works to deal with its limitations.

Response: We appreciate the reviewer’s suggestion. We have expanded the Conclusion section as follows:

<original>

Overall, our results suggest that the use of inverted images of the patient’s own nonparalyzed hand while performing MI is a useful method to increase MI vividness and cortical activity in stroke patients.

<revised>

The results of this study showed that the MI vividness and cortical activity in the paralyzed upper limb of a stroke patient were significantly higher when the subject's own hand was inverted than when no inverted image was shown. The results of this study suggest that MI vividness can be enhanced and cortical activity during MI can be activated by presenting the subject's own hand inverted image when practicing more effective MP. Our future studies will examine effective video presentation methods for practicing MP in stroke patients, using healthy subjects as a control group. The intervention effects on the paralyzed upper limb function should be evaluated when practicing MP using a video presentation method that can increase MI vividness in stroke patients.

Round 2

Reviewer 1 Report

The author has solved the problems raised previously